

# An assessment of trends and potential future changes in groundwater-baseflow drought based on catchment response times

Jost Hellwig[1], Kerstin Stahl[1]

[1]Faculty of Environment and Natural Resources, University of Freiburg, 79098 Freiburg, Germany

*Correspondence to*: Jost Hellwig (jost.hellwig@hydrology.uni-freiburg.de)

**Abstract.** Drought is an important natural hazard with large impacts on society. Changes in drought characteristics have been studied for different parts of the hydrological cycle, but insights in changes of groundwater resources are obscured due to the lack of long-term observations and large heterogeneity of hydrogeological conditions. Moreover, predicted future changes in precipitation are uncertain and have a lagged effect on streamflow and groundwater. We investigated past changes and potential

future changes in catchment baseflow as a reflexion of groundwater drought for 338 headwater catchments across Germany based on catchments' characteristic response times. First, baseflow dynamics as a proxy of groundwater storage and outflow on catchment scale were derived from streamflow records and related to precipitation input. Second, past trends in baseflow minima were calculated and attributed to climate and catchment controls. Last, response times and the timing of yearly baseflow minima were combined into scenario-neutral estimates of the sensitivity to future precipitation changes. Baseflow

response times of the studied headwaters are heterogenous across Germany, ranging from few months to several years, and depend significantly on hydrogeological conditions. Few significant trends were found in past baseflow minima and trends are highly dependent on the period of analysis. Based on the assumption of a typical regional scenario of increasing winter precipitation and decreasing summer precipitation, increases in hydrological drought hazard or no changes are projected for most parts of Germany. Catchments with longer response times can buffer interannual precipitation shifts whereas catchments

with fractured rocks are sensitive to summer precipitation decreases. These results urge for a surface- and groundwater management based on local groundwater response to precipitation and help to assess impacts of climate change on overall water supply.

## 1 Introduction

Drought is a natural phenomenon occurring in all compartments of the hydrological cycle. Accordingly, it is classified into

meteorological drought, hydrological drought, agricultural drought, socio-economic drought and groundwater drought (Mishra and Singh, 2010). Due to the large number of people affected by drought and the high economic loss related to drought events (EC, 2007) it is important to enhance the understanding of drought processes considering projected changes in drought hazard, most importantly due to climate change. Empirical analysis of monitored hydrological time series remains an important tool to validate theory-based or model-derived hypotheses on these changes, since projected future changes are often uncertain.



Particularly for the groundwater compartment there is a high diversity in response to climate input (Eltahir and Yeh, 1999; Green et al., 2011) making predictions even more difficult. Most of the empirical studies on long term trends in the drought hazard have focused on meteorological drought (e.g. Sheffield et al., 2012; Spinoni et al., 2017) and on hydrological drought (e.g. Stahl et al., 2010; Laaha et al., 2017). For groundwater drought, empirical trend analysis is difficult for two reasons: (i) groundwater time series are usually short or influenced by abstractions and (ii) where long and natural time series are available, they only give point information.

Some countries now display groundwater level anomalies at observation wells as part of their drought monitoring (e.g. Switzerland: https://www.hydrodaten.admin.ch/de/messstationen_grundwasserzustand.html; Bavaria: https://www.nid.bayern.de/grundwasser). Studies on groundwater drought mainly used borehole time series for index generation and statistical analysis (e.g. Bloomfield and Marchant, 2013; Bloomfield et al., 2015; Kumar et al., 2016; Van Loon et al., 2017; Heudorfer and Stahl, 2017; Leelaruban et al., 2017) The groundwater response to drought and its changes over time are typically very site specific and observations are therefore hardly scalable from point to catchment scale (Kumar et al., 2016). However, information on groundwater resources on the catchment scale as regional estimates are of higher relevance for effective groundwater management and adaptation to climate change. Model-based approaches for groundwater drought analysis and estimation (e.g. Peters et al., 2003; 2006; Tallaksen et al., 2009; Li and Rodell, 2015; Apurv et al., 2017) are one common way to get information on the catchment scale. There are also a few studies that analyse spring data (Fiorillo and Guadagno, 2012) or baseflow time series (Brutsaert, 2008; 2010; Fendeková and Fendek, 2012), two flux variables that provide a more integrative measure of the groundwater situation during drought.

To overcome the difficulties related to borehole data in this study we use a baseflow approach to characterize groundwater drought on a catchment scale. We analyse a large dataset of long baseflow time series in Central Europe. In this region groundwater is often used for drinking water and aquifers act as an important buffer to climatic variability. Most droughts start with a deficit in precipitation, especially when precipitation falls as rain (Van Loon and Van Lanen, 2012). For the propagation from a meteorological to a groundwater drought different processes are relevant, i.e. attenuation, delay and pooling (e.g. Peters et al., 2003; Tallaksen et al., 2009; Heudorfer and Stahl, 2017). Therefore, the drought signal in groundwater storage depends not only on current meteorological conditions, but also the previous months are important. A catchment specific time scale for this dependence may be called the catchment´s response time. Response times have been analysed by correlations between groundwater depth and time series of precipitation accumulated for different periods. Studies found, that the response times for borehole water tables (Bloomfield et al., 2015; Van Loon et al., 2017; Leelaruban et al., 2017) resp. spring discharge (Fiorillo and Guadagno, 2012) vary strongly. Moreover, some studies suggest time lags for the highest correlations between precipitation and groundwater time series because of delayed groundwater response (Bloomfield et al., 2015; Fiorillo and Guadagno, 2012). However, when looking at monthly scales this lag was always found to be quite small and often not existent (e.g. Haas and Birk, 2017).

There are two approaches to identify drought periods in a time series. The climatological approach is based on anomalies and often used also in hydrology to track the propagation of relative seasonal water deficits through the hydrological cycle (e.g.





Barker et al., 2016; Kumar et al., 2016). The traditional hydrological approach is the "threshold level approach", which defines streamflow droughts as events below a certain fixed limit and is therefore focused on actual low water availability (e.g. Yevjevich, 1967; Peters et al., 2006; Tallaksen et al., 2009). In this work we use the term drought according to the threshold-level approach, thus we consider drought events as periods of low-baseflow in absolute terms. If there is a distinct seasonal

regime, droughts mostly occur in the dry season.

Recent work on Central-European low-flows, i.e. periods when streamflow mostly consists of baseflow, found that climate change is expected to alter low-flows (Marx et al., 2018; Forzieri et al., 2014; Van Vliet et al., 2015, Gosling et al., 2017). However, the sign and magnitude of change in Central Europe is subject to model choice and emission scenario (Marx et al., 2018; Forzieri et al., 2014). Those modelling studies were focused on large river basins and the change they predicted reflects

strongly that of the precipitation change. Marx et al. (2018) found a high correlation between changes in annual precipitation sums and low-flows. Stahl et al. (2012) found that hindcasting summer low flow trends with large-scale models suggests a too homogenous spatial pattern of change compared to trends found in observations. Together with difficulties of models to capture the persistence of drought events in runoff generation found by Tallaksen and Stahl (2014) it can be assumed that some large-scale models do not necessarily resolve the heterogeneity of catchment storage and release for the hydrological response on a

headwater catchment scale. However, recent drought events demonstrated that especially headwaters are prone to groundwater-related drought impacts like shortages in water supply (Van Lanen et al., 2016). This coincides with findings that, independent from elevation, groundwater is an important catchment storage (Staudinger et al., 2017). Thus, predicting future changes in groundwater drought on catchment scale will be a prerequisite for effective drought management.

Depending on the projected climate change, different scenarios of the future development of natural baseflow can be expected

(Figure 1). If there is an increase (decrease) in precipitation projected for the entire year, flow during the dry season is also expected to increase (decrease). However, if a seasonal shift of precipitation is expected, the future development of flow during the dry season is not that straightforward. It will depend on the timing of seasonal shift and dry period and on the catchment´s characteristic response time to precipitation. Stölzle et al. (2014) found that for baseflow drought changes in precipitation are especially relevant during the recharge period which is depending on the hydrogeology of the catchment.

For many parts of Central Europe climate projections indicate a seasonal shift of precipitation to wetter winters and drier summers rather than a consistent increase/decrease (Jacob et al., 2014), urging for statistical tools to assess the prospective changes in baseflow. Knowledge on the seasonal to multi-annual scale of the baseflow response to climatic variation and extremes is therefore particularly important in Central Europe under this seasonally diverging expected climate change.

Employing a data-based approach, in this study we assess future changes in drought hazard on catchment scale across

Germany. First, past trends in baseflow drought and catchment-relevant response times are analysed. Secondly, past trends are attributed to climatic and catchment controls. Finally, based on these statistics a scenario-neutral estimate for future changes in baseflow drought is realised.





## 2 Study area and data

The dataset used in this study is the same set of headwater catchments that were used in Hellwig et al. (2018) to evaluate the representativeness of meteorological grid data. It comprises streamflow data of daily resolution available from the responsible environment agencies of the German federal states. The lengths of available time series differ, so there is a trade-off between selected period and the number of records with sufficient data. For this study, we only used catchments with streamflow data of the period 1970 to 2009. Records with data gaps were not considered. Even though the dataset consists of near-natural headwaters smaller than 200 km² and minimal regulations, records were visually screened for signs of anthropogenic influence. Four of the gauges showed spurious changes in the precipitation-streamflow relationship and were subsequently removed. The final dataset consisted of 338 gauges across Germany (Figure 2).

The selected catchments cover the flat lowland regions in the north of Germany, the low mountain ranges in south-central Germany as well as the Alps' foothills and non-glacierized front range in the south. Precipitation varies with highest annual precipitation sums in the alpine south (>2000 mm) and lowest sums in the northeast (<500 mm). Climate in Germany is humid with slightly higher precipitation sums in summer than in winter for most regions. Precipitation was analysed in the form of monthly precipitation sums taken from the European Climate Assessment and Dataset (Haylock et al., 2008), Version 13.1. According to the procedure described in Hellwig et al. (2018), catchment-specific precipitation was calculated as an area-weighted mean of the intersecting grid cells. Hellwig et al. (2018) found that due to the low spatial resolution of the meteorological dataset compared to catchment size there are some biases towards products of higher resolution, however, correlations between products were found to be very high.

Information on the hydrogeology of the catchments was taken from the digital German hydrogeological map (BGR and SGD, 2016). The catchments were classified according to the main type of porosity found in the underlying geology, either "porous" for porous aquifers such as unconsolidated alluvial fillings or "fractured" for fractured bedrock. If less than 2/3 of the catchment's area is covered by one of these types, the class "mixed" was assigned, including catchments which are dominated by other types of porosity like karst. According to this classification 80 headwater catchments (23.7 %) have mainly "porous", 170 catchments (50.3 %) mainly "fractured" and 88 catchments (26.0 %) "mixed" porosity (Figure 2).

## 3 Methods

### 3.1 Characterizing baseflow drought

#### 3.1.1 Baseflow as a measure of groundwater storage at the catchment scale

Baseflow $Q_b$ is defined as the delayed component of streamflow $Q$, i.e. flow originating from stored sources as opposed to flow originating from event water (WMO, 2008). In groundwater-dominated catchments, where groundwater is the main water storage, $Q_b$ is directly related to groundwater outflow (see Appendix A for a detailed rationale). Therefore, $Q_b$ can be taken as an integrated measure of groundwater storage $S$ as it was done in the study of Fendeková and Fendek (2012) and others. It is



important to be aware that the unknown functional relationship between $Q_b$ and $S$ strongly differs between catchments and might be non-linear, e.g. due to temporal changes in connectivity of groundwater and surface water (Elthair and Yeh, 1999; Brunner et al., 2011). Hence, without further knowledge about this functional relationship, the observation of $Q_b$ allows for conclusions about $S$ solely on an ordinal scale.

Despite the clear concept of baseflow, there is no universally valid way to separate $Q_b$ from time series of $Q$. Instead, a number of methods exist (e.g. WMO, 2008; Lyne and Hollick, 1979; Nathan and McMahon, 1990; Eckhardt, 2005; Mei and Anagnostou, 2015), that differ especially in baseflow estimation during high-flow, when $Q_b$ is relatively uncertain. For the type of extreme low-flow conditions considered in this study, most methods consistently assume that $Q$ is almost completely comprised of $Q_b$. The World Meteorological Organization recommends a baseflow separation method (WMO, 2008) that was

applied in this work using the package "lfstat" (Koffler et al., 2016) in programming language R (R Core Team, 2016). The separation procedure consists of five steps: (i) divide the time-series into non-overlapping blocks of a certain length $N$, where $N$ is recommended to be five days for rainfall regimes which have a fast streamflow response; (ii) select the minimum value for every block; (iii) compare each minimum value with the adjacent ones, if the central value is smaller $a$ times the adjacent values it becomes a turning point. $a$ is recommended to be 0.9 for rainfall regimes which have fast streamflow response; (iv)

join the turning points by straight lines; (v) compute the baseflow for every day by linearly interpolating between the turning points, if the computed value is higher than observed flow use the observation instead.

### 3.1.2 Catchment specific response times to precipitation

To quantify the catchments´ baseflow response times to precipitation ($T_R$), time series of very different characteristics need to be related to each other. To enhance comparability over space, time and different parts of the hydrological cycle, it is common

to standardize time series. Recent studies on drought either used parametric (e.g. Fiorillo and Guadagno, 2012; Barker et al., 2016) or non-parametric approaches (e.g. Bloomfield and Marchant, 2013; Kumar et al., 2016) for standardization. Parametric approaches like the Standardized Precipitation Index (SPI), introduced by McKee et al. (1993), rely on the fitting of a theoretical distribution. Since time series of precipitation ($P$) and $Q_b$ have distinct characteristics, entirely different distributions would have to be selected. To ensure consistency in the way of standardization, in this study a non-parametric approach was

applied (Bloomfield and Marchant, 2013; Kumar et al., 2016). All time series (P in accumulation periods from 1 to 36 months, $Q_b$) were standardized according to the procedure of the Standardized Groundwater Index SGI (Bloomfield and Marchant, 2013).

$P$ (resp. $Q_b$) was transformed by using the inverse normal cumulative distribution function to calculate $n$ equally spaced $p_i$ values between $1/(2n)$ and $1 - 1/(2n)$, where $n$ is the number of observations. These values were afterwards assigned to the

time series by reordering them according to the ranks of the original time series. Hence, the ranks of the original time series and the new ones coincide but the newly generated time series are normally distributed. Standardization was done separately for all 12 months of the year to account for the regime.

For all 338 catchments, correlation coefficients $R$ between $P$ and $Q_b$ were calculated,





$$R = \frac{cov(x,y)}{\sigma_x \sigma_y} = cov(x,y) \qquad (1)$$

where $x$ and $y$ are $P$ and $Q_b$, with cov() the covariance and $\sigma$ the standard deviation of each time series, that equals one for standardized time series. To derive the catchments` response times to precipitation, precipitation-accumulation periods from 1 to 36 months were tested as well as time lags of these indices to baseflow from zero to ten months. For each catchment, the combination of precipitation accumulation period and time lag, that has the highest correlation coefficient, was selected as the optimal accumulation period and the optimal time lag to describe the $Q_b$ response. Longer optimal accumulation periods indicate higher groundwater attenuation of precipitation input whereas higher time lags represent a more delayed groundwater response to precipitation. To account for seasonal variations in the response times, correlations were calculated separately for all seasons (winter = DJF, spring =MAM, summer = JJA, autumn = SON).

### 3.1.3 Catchment-relevant recharge period

Catchments analysed in this study follow a distinct regime in $Q_b$. Thus, droughts in $Q_b$ are most likely to occur in a certain period of the year. To derive the catchment characteristic drought season, annual minimal seven-day $Q_b$ (=$Q_{b7}$) were calculated. The most frequent month of $Q_{b7}$ was assigned as the month of highest baseflow drought hazard ($M_{Qb7}$). Subsequently, the derived seasonal $T_R$ was used to determine the months influencing $Q_b$ in $M_{Qb7}$. For example, if $M_{Qb7}$ in a catchment was October and $T_R$ for that catchment in autumn was six months, then the catchment-relevant recharge period would be May through October.

### 3.2 Detection of past baseflow drought trends

Time series of $Q_{b7}$ for the time period of 1970-2009 were used to analyse past trends in minimal baseflow. The analysis was carried out with the non-parametric Mann-Kendall test (MK), which is a common tool to detect monotonic increases or decreases in hydrological time series (e.g. used by Douglas et al. 2000, Lins and Slack 1999, Lorenzo-Lacruz et al. 2012, Rennermalm et al. 2010, Asarian et al., 2016) and is resistant to outliers. Calculation was done after pre-whiten the time series to account for the influence of serial correlation (see Appendix B for calculation steps).

Trends only give information on the period they are calculated for. Many studies found that trends of a certain period are not part of a trend on another timescale (e.g. Stahl et al., 2010; Hannaford and Buys, 2012; Giuntoli et al., 2013; Hannaford et al., 2013). The analysis of trends for multiple periods may help to assess whether observed trends are steady or rather fluctuating. To evaluate the trends found for the period 1970-2009, we additionally calculated trends over multiple periods for the five gauges with longest continuous records. Three of these are in the porosity-class "fractured" and one each in "porous" and "mixed".



### 3.3 Attribution of baseflow drought trends

Trends in $Q_{b7}$ were assumed to be driven by both climatic (climate control) and catchment-specific (catchment control) factors. Thus, the results were related to a set of potential predictors: (i) trends in $P$ for the catchment-specific relevant recharge period, (ii) $T_R$ for the catchment-specific $M_{Qb7}$, (iii) $M_{Qb7}$, (iv) dominant porosity of the catchment, (v) catchment size ($A$). Statistical significance of the factors was determined using the correlation coefficient $R$ (Eq. 1) for the continuous variables (trends in $P$, $A$), Spearman's rank-based correlation coefficient for the discrete variable ($T_R$,) and an analysis of variance (ANOVA) combined with the post-hoc Tukey's test (for details see Appendix C) for the nominal-scaled ones ($M_{Qb7}$, dominant porosity).

### 3.4 Scenario-neutral estimate of drought sensitivity to expected future precipitation change

Since past trends derived from empirical trend analysis (e.g. MK) are solely valid for the observation period they cannot be extrapolated beyond the period of data availability. Future drought predictions therefore mostly rely on climate projections and process modelling. For Germany climate projections indicate little to no changes of annual precipitation sums but seasonal shifts to lower summer precipitation and higher winter precipitation. However, the magnitude of the shifts differs considerably for different projections (Zebisch et al., 2005; Jacob et al., 2012; 2014; Hübener et al., 2017). A common approach to deal with such uncertainties is to use a range of possible trajectories to model hydrological change. Instead of using uncertain quantitative inputs in forward modelling, here we propose a more qualitative inverse approach. We assume that the general direction of future development is the most important information for future groundwater management planning and formulate a qualitative test scenario of the consistent direction of different projections of future precipitation change. Thus, the approach is scenario-neutral regarding emission scenarios and climate models.

Trends in future baseflow drought hazard were assumed to depend not only on precipitation changes but also on $T_R$ and $M_{Qb7}$ calculated for every catchment. A particular test scenario can still have three possible outcomes regarding future changes in baseflow drought hazard (Fig. 3): (i) increased baseflow – there are relevant months with increasing precipitation but not with decreasing; (ii) decreased baseflow – there are relevant months with decreasing precipitation but not with increasing; (iii) no change – either there are relevant months with both increasing and decreasing precipitation or there no relevant months with any change.

The test scenario applied in this study is a decrease of precipitation in summer (JJA) and an increase of precipitation in winter (DJF) with no change in the annual precipitation sum. To derive the future change in $Q_{b7}$ for a catchment we analyse whether months with increasing or decreasing precipitation belong to the catchment-relevant recharge period. For example, for a catchment with a relevant recharge period from May to October we would estimate an increased baseflow drought hazard, since there are months with decreasing precipitation in the relevant recharge period (June, July and August) but no months with increasing precipitation. This analysis was carried out for all 338 catchments.





## 4 Results

### 4.1 Past minimal baseflow trends

$M_{Qb7}$ occurs in most parts of Germany in late summer or autumn with some exceptional winter low-flow catchments in the mountainous regions (Figure 4). Seasonal response times vary across Germany, ranging from short subseasonal response times
(1-3 months) to response times of over a year (Figure 4b). Altogether 22 out of 338 catchments (~6.5 %) have a time lag in the response, though these are mainly short. All five catchments with a time lag above one month have response times longer than 12 months. According to the MK-Test, 40 out of the 338 catchments show a significant trend of $Q_{b7}$ (Figure 4c), corresponding to 11.8 %. Assuming independence of the baseflow observations, this is slightly more than the expected 5 % in case of no real trends. Hence, for the period of 1970-2009 there are some changes detectable for all catchments´ porosity
classes. For $P$ there are even less significant trends than would be expected by chance (Figure 4d). However, the results are skewed towards predominantly positive trends in $P$ during the relevant recharge periods across Germany.

The five selected gauges for a trend analysis on multiple periods all have a negative z-score in the original period 1970-2009 (Figure 5). However, in all cases also a significant positive trend could have been observed, if another period had been chosen for analysis (Figure 5). This reveals a high influence of the observation period on the results of the trend analysis. The high
variability of trends is also independent of the porosity, $T_R$ and $M_{Qb7}$.

### 4.2 Trend attribution

None of the factors tested was found to explain past trends of $Q_{b7}$ very well. This lack of statistical relation coincides with the results that few trends are significant and trend magnitude is highly dependent on the period of analysis. However, there is a small but significant correlation between the trends in $P$ for the catchment-relevant recharge period and trends in $Q_{b7}$ (Figure
6). Moreover, the ANOVA indicates a significant influence of $M_{Qb7}$ on the trends in $Q_{b7}$ even though the Tukey's test does not distinguish two groups significantly. The largest differences are between July and November, so the season of low-flow might be relevant for trend magnitude in $Q_{b7}$ as well. The remaining factors ($T_R$, dominant porosity and $A$) do not show any relation to the trends in $Q_{b7}$ (Figure 6).

### 4.3 Potential future drought hazard

According to the test scenario, $Q_{b7}$ will decrease or not change for most of Germany (Figure 7). The only catchments with estimated increases of $Q_{b7}$ are located in the mountainous regions of south-east Germany, especially in the Alpine foothills in the catchments with annual low-flows in winter.

The changes in drought hazard are significantly related to the catchments' response times $T_R$ (Figure 8). Moreover, there is a strong statistical linkage to the dominant porosity (Figure 8b). Catchments dominated by fractured rocks more frequently show
estimated decreases in $Q_{b7}$ than the other classes. This coincides with a significant relationship between $T_R$ and the dominant porosity of the catchment (Figure 8c). According to Turkey's test catchments with fractured rocks have shorter response times



than the other classes. Contrary, catchments with porous aquifers or mixed hydrogeology, which have long $T_R$, can more often compensate summer precipitation decreases with winter increases and are therefore more frequently in the category "no change".

## 5 Discussion

### 5.1 The catchment baseflow response to precipitation

Catchments´ response times to precipitation were found to be highly diverse across Germany ranging from one month to three years. In general, baseflow response times determined as a proxy of groundwater response are rather short compared to other studies. Fiorillo and Guadagno (2012) found response times of 12 to 24 months for a karst region in southern Italy, and for shorter precipitation accumulation periods highest correlations when adding a short time lag. Marchant and Bloomfield (2013) also found in 3 out of 14 cases a time lag for highest correlations. We found time lags to be an exception, supporting the results of Kumar et al. (2016), Barker et al. (2016) and Haas and Birk (2017). This indicates that in the headwater catchments studied, the delay of the groundwater baseflow response to meteorological conditions may be shorter than one month and therefore not detectable on the monthly scale whereas the attenuation of meteorological variability is clearly attributable to characteristic precipitation accumulation periods.

The large differences of baseflow response times for different porosity classes match the theoretical assumptions that baseflow strongly depends on hydrogeological conditions. For the entire streamflow, differences were found to be much smaller (not shown, compare e.g. Haslinger et al., 2014), since other processes like overland flow are also important. The patterns of $T_R$ which are related to the catchments' hydrogeology support the assumption that baseflow can be used as a proxy of the groundwater situation on catchment scale.

Consistent with the work by Bloomfield et al. (2015), we found that hydrogeology is a highly relevant factor for the explanation of different groundwater baseflow response times. Kumar et al. (2016) did not find a relationship between the hydraulic conductivity and groundwater response time for boreholes. A possible reason for this different finding is that for point data even small local influences, that are hard to determine, are quite relevant (e.g. human influences), whereas baseflow reflects more the overall situation within the catchment. Small influences may be negligible at this scale and the underlying influence of hydrogeology easier to detect.

In general, the results indicate that groundwater storage – represented by baseflow – is on catchment scale vitally driven by precipitation in the relevant recharge period. However, the season of low-flow is also expected to have an influence: regimes with winter low-flows in Central Europe are governed by snow storage during that season. Thus, not only precipitation but also temperature is a major factor for that catchments. Moreover, it is impossible to distinguish snow melt from groundwater outflow during baseflow separation. Therefore, a baseflow approach does not allow for conclusions on groundwater storage in snow-dominated catchments.



## 5.2 Past and potential future changes in low baseflow

The trend analysis revealed, that the trend in baseflow minima is highly depending on the period it is calculated for. The observation of a trend in $Q_{b7}$ thus is a poor indicator of future developments. Despite the relatively narrow seasonality of low baseflow timing (except for the alpine areas), trends are highly variable across the region - a result of the likewise variable
response times. However, the correlation tests for attribution of the trends revealed, that precipitation during the catchment-relevant recharge period is an important factor, which confirms previous model-based findings for part of the region (Stölzle et al., 2014). Thus, this catchment-relevant recharge period should be considered for projections.

The method employed in this study provides a straightforward projection of the probable future directions of changes in baseflow resp. groundwater drought under climate change. We accounted for the uncertainty in future climate projections by
taking only concordant directions of precipitation change in our scenario-neutral approach. Contrary to future climate projections, past trends in precipitation for the catchment-relevant recharge period were found to be small and mostly positive. Similarly, Kopp et al. (2018) found for southern Germany a high variability of annual groundwater recharge without distinct trends. In the past precipitation trends have not been seasonally diverging, but climate models suggest that changes will become more relevant in the second half of the century (e.g. Jacob et al., 2012). As the magnitude of the trends differs for the climate
models, we did not quantify our scenario and thus did not quantify the magnitude of future baseflow as well.

Catchments' characteristic response times were assumed to remain constant. Under a more extreme climate change however, changes in catchments´ responses (e.g. due to non-stationary response times) cannot be excluded. Based on assumed precipitation change in the catchments' respective recharge periods, decreasing $Q_{b7}$ (i.e. increased groundwater drought hazard) were estimated for the majority of the catchments. This is because of the timing of $M_{Qb7}$, which occurs predominantly in
summer/autumn. Only those catchments with long response times can compensate the decreases in summer precipitation and only in the Alpine foothills, where $M_{Qb7}$ is in winter, a decrease in drought hazard is predicted. However, in this region winter precipitation is mainly snow and therefore does not immediately contribute to groundwater recharge. Moreover, air temperature and its changes over time – which are not considered in the test scenario – are especially important for winter low-flows. Hence, uncertainty is quite high for these projections.

## 6 Conclusions

Climate change is expected to alter the hydrological drought hazard. However, uncertainty of climate projections and even contrasting seasonal changes impede a straightforward assessment of the prospective changes in Central Europe and elsewhere. Here we presented a statistical approach to estimate the potential direction of future changes in hydrological drought hazard. Past trends were found too variable to provide a consistent regional picture of past and expected changes because of their high
dependency on the trend calculation period. But they did allow to test the attribution of trends in baseflow to precipitation changes in catchment specific recharge periods. Based on that information, a more process-oriented approach was developed, using the catchments´ characteristic response times to precipitation and the relevant recharge periods for scenario-neutral





projections. Scenario-neutral projections are efficient alternatives to ensemble projections and target the most important information for management. Especially for regions where directions of climate change are seasonally varying they can provide valuable insights into the basic changes of the system.

Catchments with short response times were found to have a high probability for a decrease in baseflow minima and hence an
increase in the groundwater drought hazard, as seasonal changes cannot compensate each other. However, there is no homogeneous pattern of response times across central Europe and so predicted changes of groundwater drought hazard are also regionally varying. This urges for a regionally adapted groundwater management based on the local catchment response times. As past events like the 2015 central European drought already caused groundwater related drought impacts in headwater regions, there is an urgent need for adaptation in catchments facing even higher drought hazard in the future.

The diversity of response times, few long-time data on groundwater storage and the absence of distinct past trends in precipitation and hydrological variables limit the potential to generalize the results. On the way to extensive predictions of future groundwater drought hazard across Central Europe, further model-based work will be needed. Reproducing the catchment relevant response times with high-resolution large-scale models may be key for an assessment of future changes and related implications for groundwater management under various scenarios and for ungauged catchments.

**Data availability**

Streamflow data are on request available for scientific purposes from the responsible federal state agencies, i.e. the State Environmental Agency of Baden-Württemberg (https://www.lubw.baden-wuerttemberg.de/), Bavaria (http://www.gkd.bayern.de/), Brandenburg (http://www.lfu.brandenburg.de/), Hesse (https://www.hlnug.de/), Mecklenburg-Western Pomerania (https://www.lung.mv-regierung.de/), Lower Saxony (https://www.nlwkn.niedersachsen.de/), North
Rhine-Westphalia (https://www.lanuv.nrw.de/), Rhineland-Palatinate (https://lfu.rlp.de/), Saarland (https://www.saarland.de/landesamt_umwelt_arbeitsschutz.htm), Saxony (https://www.smul.sachsen.de/lfulg/), Saxony-Anhalt (https://lhw.sachsen-anhalt.de/), Schleswig-Holstein (http://www.schleswig-holstein.de/DE/Landesregierung/LKN/lkn_node.html) and Thuringia (http://www.thueringen.de/th8/tlug/). Climate data are available via the website of the European Climate Assessment & Dataset (http://www.ecad.eu/), the German hydrogeological
map (BGR & SGD, 2016) is online available as well (https://www.bgr.bund.de/DE/Themen/Wasser/Projekte/laufend/Beratung/Huek200/huek200_projektbeschr.html).



## Appendix A: Mathematical relationship between baseflow and groundwater storage

In groundwater dominated catchments $Q_b$ is an integrated signal of groundwater conditions in the entire area and mainly depends as a function $g$ on the hydraulic head $H$ of the groundwater:

$$Q_b = g(H) + r \text{ with } \frac{dQ_b}{dH} > 0 \; \forall \, H \tag{A1}$$

where $r$ is the additional flow from other stored sources. Thereby, $H$ is a monotonic function $f$ of groundwater storage $S$:

$$H = f(S) \text{ with } \frac{dH}{dS} > 0 \; \forall \, S \tag{A2}$$

Combining Eq. (A1) and (A2) leads to

$$Q_b = g(f(S)) + r = k(S) + r \text{ with } \frac{dQ_b}{dS} > 0 \; \forall \, S \tag{A3}$$

Because of the monotonic behaviour, function $k$ is reversible:

$$S = k^{-1}(Q_b - r) \text{ with } \frac{dS}{d(Q_b - r)} > 0 \; \forall \, Q_b - r \tag{A4}$$

Potential sources for $r$ include snow melt, interflow, lake outflow, sewage discharge or other anthropogenic sources of discharge. Anthropogenic influences can be assumed negligible, as catchments in this study were selected as near-natural. Moreover, there are no big lakes or other surface water storages in the catchments. Snow melt remains an important factor that might blur the true signal of groundwater storage. This is particularly important for the catchments in higher elevations, e.g. in the Alps, and must be considered for the interpretation of the results. However, in catchments where these factors are of minor relevance compared to groundwater outflow, Eq. (A4) simplifies to

$$S \simeq k^{-1}(Q_b) \text{ with } \frac{dS}{d(Q_b)} > 0 \; \forall \, Q_b \tag{A5}$$

i.e. higher baseflow indicates higher groundwater storage and vice versa. However, without further knowledge about $k^{-1}$, the observation of $Q_b$ allows for conclusions about $S$ solely on an ordinal scale.

## Appendix B: The Mann-Kendall Trend Test with pre-whitening

To calculate trends, the non-parametric Mann-Kendall-Test (MK) was applied. However, the results are affected by serial correlation which increases the Type I error (i.e. reject the no trend hypothesis although there is no trend). To test for serial correlation in the data we fitted an autocorrelation AR(1) model to each $Q_{b7}$ time series. The calculated autocorrelation is significant on a level of $\alpha = 0.05$ if absolute autocorrelation is higher than $1.96/\sqrt{n}$ (Douglas et al., 2000) where $n$ is the length of the time series (in this case $n = 40$). 130 out of 338 time series showed significant serial correlation, thus requiring a pre-processing before using MK. Kulkarni and Von Storch (1995) recommended to pre-whiten time series to allow for the MK. Since the sample size is relatively small here and trend magnitude not too large, pre-whiten is not expected to reduce test power of MK much in this case, but to reduce the Type I error (Bayazit and Önöz, 2007). Pre-whitening was done according to other studies (Kulkarni and Von Storch, 1995; Douglas et al., 2000) as in Eq. (A6):

$$Y_t = X_t - r_1 \, X_{t-1} \tag{A6}$$





where $Y_t$ is the pre-whitened time series at time step $t$, $X_t$ is the original time series at time step $t$ and $r_1$ is the serial correlation determined with the AR(1) model. Pre-whitening reduced the serial correlation in all time series to close to zero.

The MK-Test compares the number of concordant pairs in the data with the number of discordant pairs. This gives the Kendall score $S_K$ (Kendall, 1948)

$$S_K = \sum_{j>i} sign(Y_j - Y_i)\, sign(Z_j - Z_i) \tag{A7}$$

where sign() is a function that returns the algebraic sign. For the Mann-Kendall trend test, that was applied here, $Z$ is the time. Therefore, the second part of Eq. (A7) always equals +1 and it simplifies to

$$S_K = \sum_{j>i} sign(Y_j - Y_i) \tag{A8}$$

In case of no real trend $S_K$ has a mean value of zero and in case of no ties in the data a standard deviation $\sigma_S$ of (Kendall, 1948)

$$\sigma_S = \sqrt{\frac{n(n-1)(2n+5)}{18}} \tag{A9}$$

Thus, $S_K$ can be transformed into the standard normal distributed z-score by dividing by the standard deviation of $S_K$:

$$z = \begin{cases} \frac{S_K - 1}{\sigma_S} & if\ S_K > 0 \\ 0 & if\ S_K = 0 \\ \frac{S_K + 1}{\sigma_S} & if\ S_K < 0 \end{cases} \tag{A10}$$

$z$ was assumed to be significant if it was not within the interval $\pm 1.96$, i.e. a significance level of $\alpha = 0.05$.

## Appendix C: ANOVA and Tukey's test

To detect the influence of a categorical variable with multiple levels on a numerical variable an ANOVA was used in this work. The ANOVA compares $S_e$, the sum of squares explained by the categorical variable´s levels, with the sum of squares of the residuals ($S_r$). The mean sums ($M_{Se}$, $M_{Sr}$) are tested for significance using the F-test

$$F = \frac{M_{Se}}{M_{Sr}} = \frac{S_e/d_{fe}}{S_r/d_{fr}} \tag{A11}$$

where $d_f$ denotes the degrees of freedom ($d_{fe}$: number of levels minus one, $d_{fr}$: amount of data minus $d_{fe}$ minus one). The test statistic $F$ follows the F-distribution and was assumed to be significant for $p<0.05$.

The ANOVA gives information about the general significance of categorical variable. An equally important information is which of the categorical variable's levels differ significantly regarding the target variable. This information was obtained using the post-hoc Tukey's test. For this analysis a pairwise t-test between all levels is carried out

$$t = \frac{\bar{x}_1 - \bar{x}_2}{\sqrt{\frac{s_1^2}{n_1} - \frac{s_2^2}{n_2}}} \tag{A12}$$

where $\bar{x}_i$ and $s_i^2$ are the mean resp. the variance for the two levels, and $n_i$ is the amount of data per level. The Tukey's test accounts for the multiple comparison problem (i.e. a higher probability of getting significant results by chance due to a higher



number of comparisons) by using the studentised range distribution instead of the t-distribution. Again, the results were assumed to be significant for p<0.05.

## Acknowledgements

We acknowledge the data providers at the federal states´ agencies for the streamflow data, the E-OBS dataset from the EU-FP6 project ENSEMBLES (http://ensembles-eu.metoffice.com) and the data providers in the ECA&D project (http://www.ecad.eu). The authors were funded by the DFG (JH: project TrenDHy DFG STA632/4-1, KS: Heisenberg programme STA632/3-1). Benedikt Heudorfer provided helpful comments on the manuscript.

## Competing interests

The authors declare that they have no conflict of interest.

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





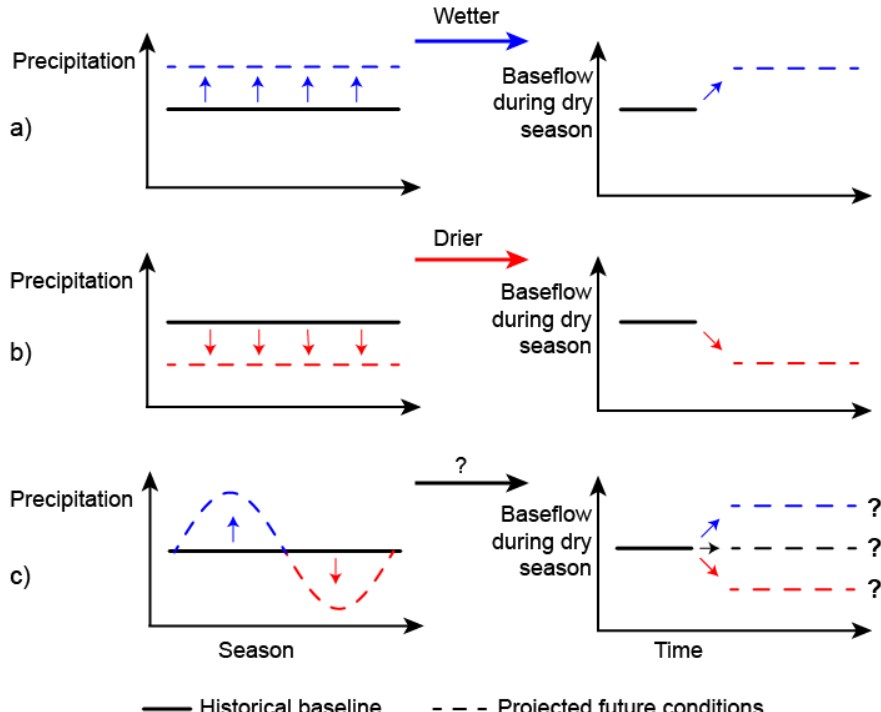

**Figure 1: Schematic direction of change of natural baseflow in the dry season under different climate change scenarios: a) all seasons become wetter, b) all seasons become drier & c) seasonal shift of precipitation.**





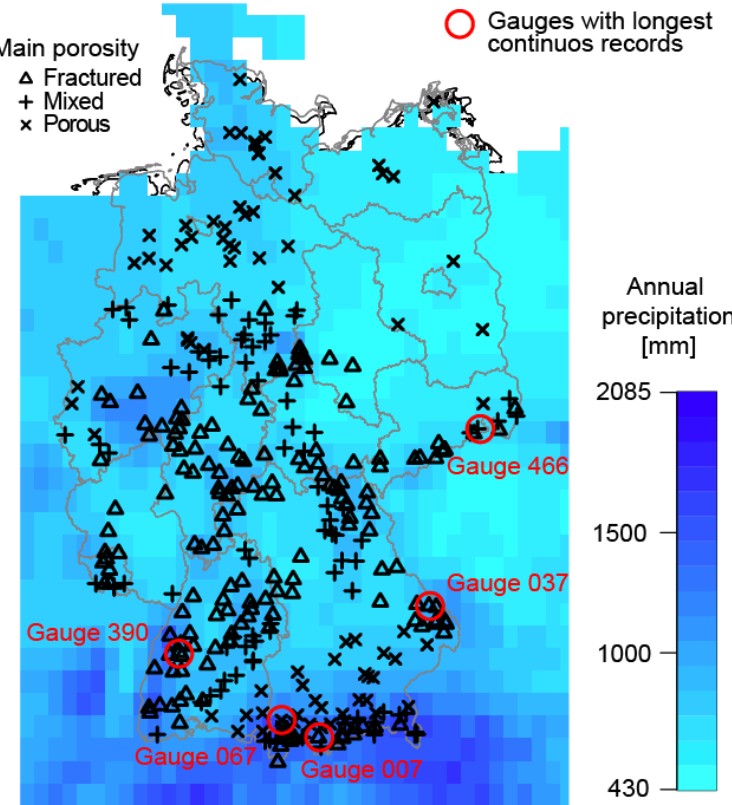

**Fig. 2: Location of gauges in Germany, catchments' dominant type of porosity (derived from the German hydrogeological map) and mean annual precipitation sums (calculated from European Climate Assessment and Dataset E-OBS).**





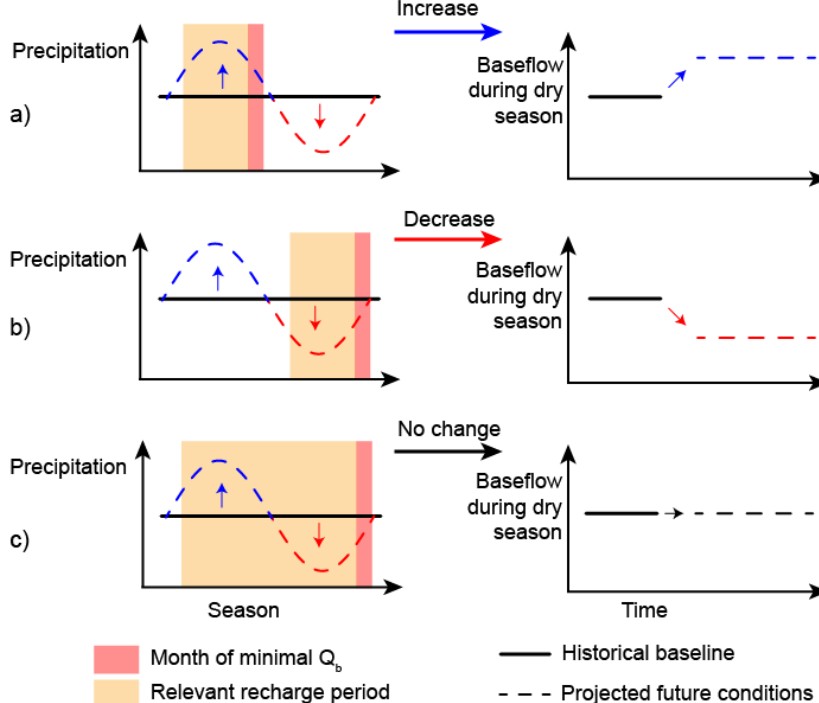

Fig. 3: Potential directional changes in $Q_{b7}$ under a constant seasonally diverging precipitation change scenario, depending on $T_R$ and $M_{Qb7}$: a) increase, b) decrease & c) no change.



**Fig. 4: Patterns of $Q_{b7}$ across Germany. a) low-flow month $M_{Qb7}$ and b) seasonal response time $T_R$. c) Trend of $Q_{b7}$ and d) $P$ in the relevant recharge period for 1970-2009.**





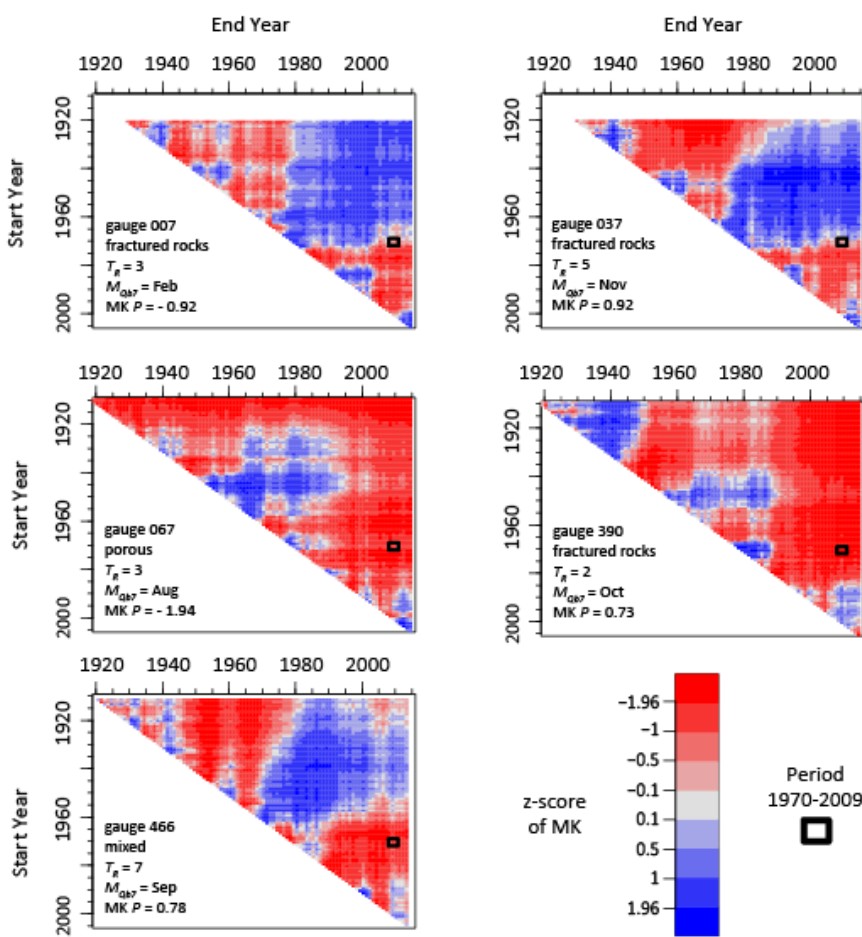

Fig. 5: z-score values for all possible periods for the five catchments with longest continuous records.





**Fig. 6: Attribution of trends in $Q_{b7}$ to potential influencing factors. a) trends in precipitation $P$ during the relevant recharge period; b) length of response time $T_R$; c) $M_{Qb7}$; d) dominant porosity in the catchment and e) catchment size $A$. Boxes represent the quartiles 0.25 and 0.75 and whiskers last to the most extreme value within a maximum distance of a 1.5-fold interquartile range.**





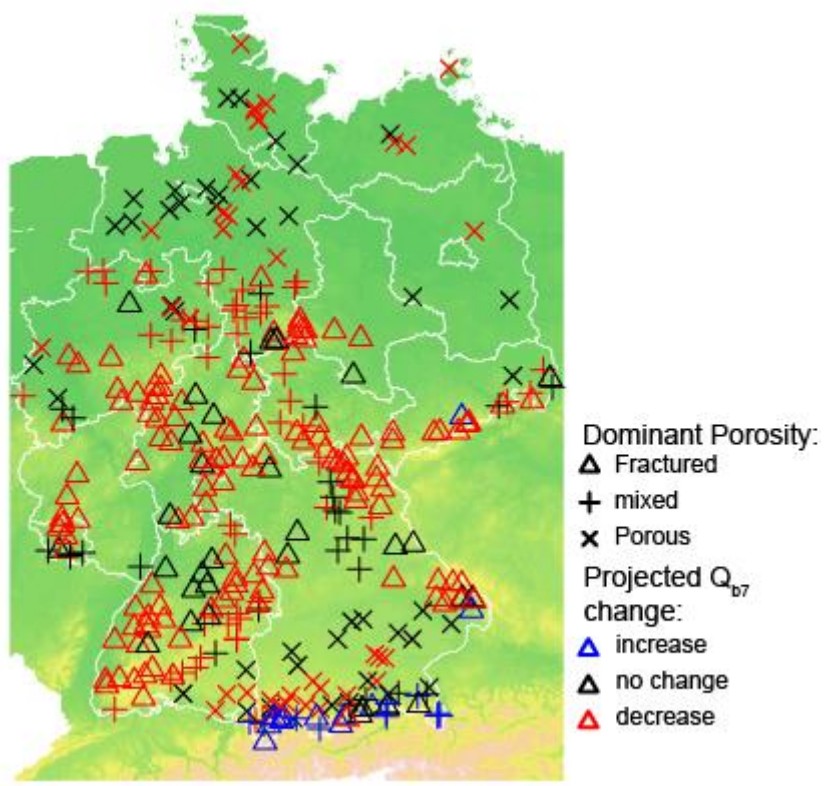

Fig. 7: Future changes in $Q_{b7}$ according to the test scenario.





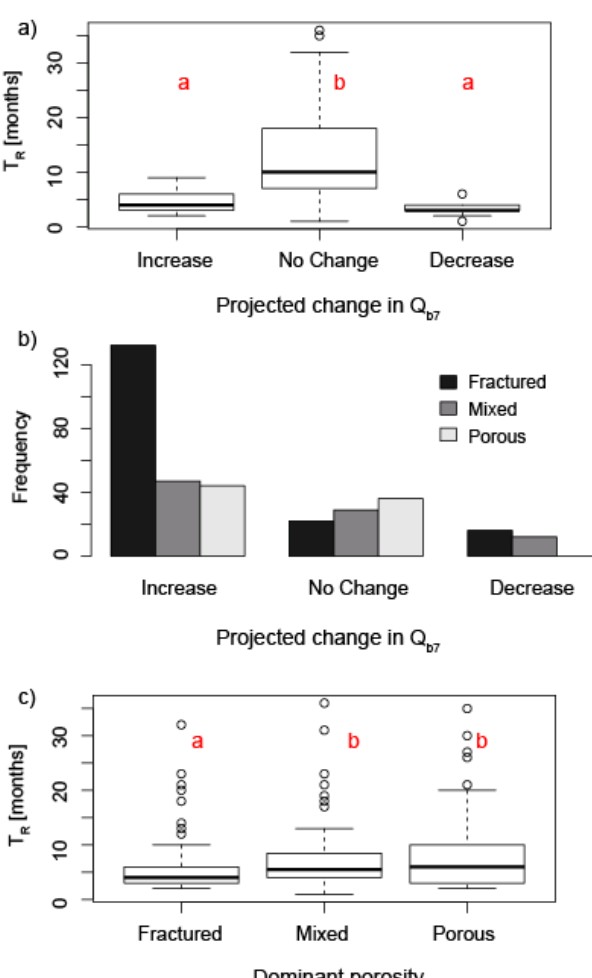

**Fig. 8: Relationships of projected changes, $T_R$ and catchment`s dominant porosity. a) Distribution of $T_R$ for projected changes; b) frequency of catchments with a certain dominant porosity for different projected changes and c) distribution of $T_R$ for dominant porosities. Boxes represent the quartiles 0.25 and 0.75 and whiskers last to the most extreme value within a maximum distance of a 1.5-fold interquartile range. Red letters indicate results of the Tukey´s post-hoc test to distinguish significantly different groups.**