# Peer review of "An assessment of trends and potential future changes in groundwaterbaseflow drought based on catchment response times"

_Hydrology and Earth System Sciences, 2018_

## Short Comment (SC1) · 18 Jun 2018

**General Comments**

This is a sound and timely paper that builds on established methods and previous work of others and examines important water resource issues with management implications in the face of climate change. There is a thorough introduction and discussion of methods. The analysis is based on a very large sample size of existing gages which strengthens the validity of the results and conclusions. Methods used in the analysis are well documented and established procedures in the field. There is a detailed appendix and appropriate and easily interpretable figures. I particularly appreciated figure 5, a very compelling illustration of the impacts of start and end year on trend analyses.

Specific Comments

The study includes data from presumably a wide range of catchment types based on the spatial distribution of gages. However, the only catchment attribute discussed is a measure of porosity. I would have preferred some greater detail of catchment geology, land use/vegetation, and geomorphic characteristics to differentiate the study basins.

I don't fully understand the use of the "scenario-neutral" term. Although emissions specific scenarios are not discussed, only one trajectory is analyzed (reduced summer precip. and increase winter precip.). It would be interesting to test the baseflow impacts of varied changes in seasonal precipitation pattern.

How are summer low flows in the Alpine foothills projected to change? Although minimum flows currently occur in winter, low flows in summer can have greater water resources and ecological implications.

It was important to note in section 5.2 that: "Catchments' characteristics response times were assumed to remain constant". Although likely outside the scope of this article, it would be nice to explore this more and the implications on baseflow recession.

Technical Corrections

5.2, line 1 should be dependent rather than depending.

This is an easily readable and well formatted text.

---

## Referee Comment (RC1) · J.P. Bloomfield (Referee) · 26 Jun 2018

The paper presents an analysis of baseflow 'to characterize groundwater drought on a catchment scale'. Trends in observed baseflow minima and derived drought descriptors are identified and investigated in the context of 'climatic and catchment controls'. A 'scenario-neutral' approach is adopted to characterise the sensitivity of the drought descriptors to future changes in. The study uses flow data from 338 gauges in headwater catchments across Germany.

[Figure]

General comments

Introduction:

The aims of the study could be set out more clearly in the Introduction. The two following statements 'we use a baseflow approach to characterize groundwater drought on a catchment scale' (p2, l19) and 'Employing a data-based approach, in this study we assess future changes in drought hazard on catchment scale across Germany' (p3, l29) describe what has been done, but there is no unambiguous aim or research question stated in the Introduction.

Study area and data:

Data from 338 gauges on headwater catchments were used in this study. In this context, what constitutes a headwater catchment?

It's not easy to tell from Figure 2, but it is possible that some of the gauges are nested catchments. Is this the case? If so, what biases if any might this introduce into the data? What are the implications of those potential biases for the trend analysis (section 4.2) and the results of the potential future drought hazard assessment (section 4.3)?

Streamflow data for the period 1970-2009 was analysed. Was the data complete? If there was any missing data how was it accounted for in the pre-processing of the data?

The data 'were visually screened for signs of anthropogenic influence' and 'four of the gauges showed spurious changes . . . and were subsequently removed' (p4, l7-8). Please could you justify their exclusion in more detail. What anomalies were present in the data that caused you to exclude the sites?

Why was a 2/3rds fraction used to define mixed catchment (p4, l21)? Is there a citation for this?

Is there any information on the distribution of low permeability superficial deposits across the region? It's not uncommon for such deposits to play an important role in

stream flow generation and so the proportion of such deposits in catchments might be an interesting parameter to investigate in the context of the study, and should at least be commented on, either here or in the Discussion.

Methods:

The justification for the use of Mann-Kendall (MK) test could be more robust. There is a significant literature on the application of this test to hydroclimatic time series. However, it's application to such time series is also contentious when there are underlying auto and cross-correlations present. Consider adding to the justification of use of the z-statistic [also see comment below about identification of significant trends based on MK test].

Results:

Figure 4c is a map of the MK z-statistic indicating the direction and magnitude of the trend in Qb7. However, it is stated that 'according to the MK-Test, 40 out of the 338 catchments show a significant trend in Qb7' (p8, l7). What was the level of signifi-cance that was used? Please justify this and link this justification back to the Methods, section3.2?

Discussion:

It is noted in Section 5.1 of the Discussion that 'a baseflow approach does not allow for conclusions on groundwater storage in snow-dominated catchments' (p9, l30). In this context, how do you define snow-dominated catchments, and which if any of the 338 headwater catchments that you have analysed fall into this category? If any of the catchments are 'snow-dominated' should they be excluded from your analysis?

Specific comments

P1, l10: replace reflexion with reflection

P2, l28: resp. is not a normal abbreviation to use in articles like this. If it is and

abbreviation for respectively please re-write [see also p5, l28, p10, l9 and p13, l25]

P9, l9, replace Marchant and Bloomfield (2013) with Bloomfield and Marchant (2013)

P10, l5, delete comma after revealed

---

## Author Comment (AC1) · 23 Aug 2018

We would like to thank Spencer Sawaske for his feedback and constructive suggestions on this manuscript. We think that revisions based on the reviewers' comments will further improve our work. Below we give point-by-point responses to the comments (blue and italic).

*General Comments*

*This is a sound and timely paper that builds on established methods and previous work of others and examines important water resource issues with management implications in the face of climate change. There is a thorough introduction and discussion of methods. The analysis is based on a very large sample size of existing gages which strengthens the validity of the results and conclusions. Methods used in the analysis are well documented and established procedures in the field. There is a detailed appendix and appropriate and easily interpretable figures. I particularly appreciated figure 5, a very compelling illustration of the impacts of start and end year on trend analyses.*

*Specific Comments*
*The study includes data from presumably a wide range of catchment types based on the spatial distribution of gages. However, the only catchment attribute discussed is a measure of porosity. I would have preferred some greater detail of catchment geology, land use/vegetation, and geomorphic characteristics to differentiate the study basins.*
Indeed, there is a larger number of potential controls that influence hydrologic processes on catchment scale. Here we chose those factors that are most likely to influence long term changes in minimal baseflow. We differentiated only three hydrogeological classes to maintain a sufficient sample size and this appeared the most straightforward classification to explore first-order controls. There is of course more lithological diversity across Germany. One possible predictor would be the hydraulic conductivity. There are some estimates covering the whole study area, but uncertainty is very high. For example, the estimates of GLHYMPS (Gleeson et al., 2014) and HÜK200 (BGR and SGD, 2016) differ up to six orders of magnitudes for many parts of Germany. Classifying more detailed meaningful categories with enough samples each, would be a major ground work required and thus beyond the scope of this paper. We suggest to discuss this issue in more detail, but currently see no easy alternative to be used. However, for the revisions we will add land use categories and a terrain-based measure as further characteristics to differentiate study basins.

*I don't fully understand the use of the "scenario-neutral" term. Although emissions specific scenarios are not discussed, only one trajectory is analyzed (reduced summer precip. and increase winter precip.). It would be interesting to test the baseflow impacts of varied changes in seasonal precipitation pattern.*
We agree that other scenarios could be tested easily. Instead we opted to test only a generalized probable future change, as all projections agree on this general scenario (reduced summer precipitation and increased winter precipitation). The term scenario-neutral was branded first by Prudhomme et al. (2010) for tests that do not follow strict climate model runs, but we agree that we deviate from their more substantial sensitivity test/response surfaces and it may hence be better to change the terminology. However, as our test scenario is independent from the magnitude of change which depends on the model and emission scenario, the analysis is in a way "scenario neutral".
It is not in the scope of our study to identify changes due to theoretical combinations of precipitation changes but to predict the changes under certain assumptions that are common to all scenarios. Therefore, we decided not to include further scenarios.

*How are summer low flows in the Alpine foothills projected to change? Although minimum flows currently occur in winter, low flows in summer can have greater water resources and ecological implications.*

This is an important point and we will include the implications for summer low-flows in this area in the revised version. Due to the short response times many catchments in the Alpine foothills are likely to have reduced summer low-flows under the test scenario.

*It was important to note in section 5.2 that: "Catchments' characteristics response times were assumed to remain constant". Although likely outside the scope of this article, it would be nice to explore this more and the implications on baseflow recession.*

For climate change impact studies, the choice of boundary conditions is often critical. Catchment characteristics (e.g. response times) can generally be assumed to be independent from climate. However, recent studies found that over a longer term these catchment characteristics are also related to climate and so climate change might impact them as well (e.g. Troch et al., 2015; Saft et al., 2016). This is often referred to as catchment coevolution. Unfortunately, complexity of these processes is high and therefore not easily predictable for a large range of conditions like they occur in Germany. Therefore, we assume constant response times but agree, that this would be an interesting topic for further research and suggest to include it more in the discussion.

*Technical Corrections*
*5.2, line 1 should be dependent rather than depending.*
Will be corrected.

*This is an easily readable and well formatted text..*

**References:**

BGR and SGD: Bundesanstalt für Geowissenschaften und Rohstoffe and Staatliche Geologische Dienste (2016), Hydrogeologische Übersichtskarte von Deutschland 1:200.000, Oberer Grundwasserleiter (HÜK200 OGWL). Digitaler Datenbestand, Version 3.0. – Hannover.

Gleeson, T., N. Moosdorf, J. Hartmann, and L. P. H. van Beek (2014), A glimpse beneath earth's surface: Global HYdrogeology MaPS (GLHYMPS) of permeability and porosity, Geophys. Res. Lett., 41, 3891–3898, doi:10.1002/2014GL059856.

Prudhomme, C., R.L. Wilby, S. Crooks, A.L. Kay, N.S. Reynard (2010), Scenario-neutral approach to climate change impact studies: Application to flood risk, J Hydrol, 390, 198-209, doi:10.1016/j.jhydrol.2010.06.043

Saft, M., M. C. Peel, A. W. Western, and L. Zhang (2016), Predicting shifts in rainfall-runoff partitioning during multiyear drought: Roles of dry period and catchment characteristics, Water Resour. Res., 52, 9290‑9305, doi:10.1002/2016WR019525.

Troch, P. A., T. Lahmers, A. Meira, R. Mukherjee, J. W. Pedersen, T. Roy, and R. Valdés-Pineda (2015), Catchment coevolution: A useful framework for improving predictions of hydrological change?, Water Resour. Res., 51, 4903–4922, doi:10.1002/2015WR017032.

---

## Author Comment (AC2) · 23 Aug 2018

We would like to thank John Bloomfield for his feedback and constructive suggestions on this manuscript. We think that revisions based on the reviewers' comments will further improve our work. Below we give point-by-point responses to the comments (blue and italic).

*The paper presents an analysis of baseflow 'to characterize groundwater drought on a catchment scale'. Trends in observed baseflow minima and derived drought descriptors are identified and investigated in the context of 'climatic and catchment controls'. A 'scenario-neutral' approach is adopted to characterise the sensitivity of the drought descriptors to future changes in. The study uses flow data from 338 gauges in headwater catchments across Germany.*

*General comments*
*Introduction:*
*The aims of the study could be set out more clearly in the Introduction. The two following statements 'we use a baseflow approach to characterize groundwater drought on a catchment scale' (p2, l19) and 'Employing a data-based approach, in this study we assess future changes in drought hazard on catchment scale across Germany' (p3, l29) describe what has been done, but there is no unambiguous aim or research question stated in the Introduction.*

Thanks for pointing this out. We agree and will set the main aim and the more specific objectives more clearly in the revised version. The main aim may perhaps read:
*This study aims to develop a data-based approach to estimate the potential direction of future changes in hydrological drought hazard under climate projection uncertainty.*

*Study area and data:*
*Data from 338 gauges on headwater catchments were used in this study. In this context, what constitutes a headwater catchment?*
The term "headwater catchment" was chosen as all catchment areas are below 200 km². We will state this more clearly in the revised version.

*It's not easy to tell from Figure 2, but it is possible that some of the gauges are nested catchments. Is this the case? If so, what biases if any might this introduce into the data? What are the implications of those potential biases for the trend analysis (section 4.2) and the results of the potential future drought hazard assessment (section 4.3)?*
We agree that nested catchments would introduce biases and hence did not use any nested catchments in this study. We will add this information in the revised version.

*Streamflow data for the period 1970-2009 was analysed. Was the data complete? If there was any missing data how was it accounted for in the pre-processing of the data?*
As we report in section 2, any records with data gaps were not considered for this study.

*The data 'were visually screened for signs of anthropogenic influence' and 'four of the gauges showed spurious changes … and were subsequently removed' (p4, l7-8). Please could you justify their exclusion in more detail. What anomalies were present in the data that caused you to exclude the sites?*
For all records, summed precipitation was plotted over summed streamflow for the entire period. Under constant conditions, the plot is expected to show seasonal varying slopes but no sudden knee. However, four of the gauges had a knee, indicating some kind of anthropogenic influence. In a Master thesis, L. Gerke was able to relate the knee in the plot of one catchment to a change of the gauges' location. We will include these details in the revised version.

*Why was a 2/3rds fraction used to define mixed catchment (p4, l21)? Is there a citation for this?*

The fraction is arbitrary and was chosen to get the relatively homogenous groups "fractured" and "porous". Due to the high heterogeneity of the catchments we think it is hard to define more detailed groups with sufficient sample size. Using another reasonable threshold is not expected to change the results as the groups all cover a broad range of values. However, significance of the results could slightly change. We suggest to test this on a few examples in the course of the revision process and then decide what needs to be included based on the results.

*Is there any information on the distribution of low permeability superficial deposits across the region? It's not uncommon for such deposits to play an important role in stream flow generation and so the proportion of such deposits in catchments might be an interesting parameter to investigate in the context of the study, and should at least be commented on, either here or in the Discussion.*
We don't know of data on superficial deposits across the region that are detailed enough to use them as predictors. However, we will explore this further and in lack of data add this point to the discussion in the revised manuscript.

*Methods:*
*The justification for the use of Mann-Kendall (MK) test could be more robust. There is a significant literature on the application of this test to hydroclimatic time series. However, it's application to such time series is also contentious when there are underlying auto and cross-correlations present. Consider adding to the justification of use of the zstatistic [also see comment below about identification of significant trends based on MK test].*
*Results:*
*Figure 4c is a map of the MK z-statistic indicating the direction and magnitude of the trend in Qb7. However, it is stated that 'according to the MK-Test, 40 out of the 338 catchments show a significant trend in Qb7' (p8, l7). What was the level of significance that was used? Please justify this and link this justification back to the Methods, section3.2?*
The Mann-Kendall test is a common test for statistical analysis of time series in hydrology. The detailed description of the method applied is given in Appendix B of the manuscript. Since MK overestimates significance for autocorrelated time series, we correct in this study for serial correlation using pre-whitening (see equation A6 in the manuscript). As we think that the detailed procedure is only interesting for part of the readers, we prefer to leave most of this information in the Appendix, however we agree that the significance level ($\alpha=0.05$) may be mentioned in the main text as well.

*Discussion:*
*It is noted in Section 5.1 of the Discussion that 'a baseflow approach does not allow for conclusions on groundwater storage in snow-dominated catchments' (p9, l30). In this context, how do you define snow-dominated catchments, and which if any of the 338 headwater catchments that you have analysed fall into this category? If any of the catchments are 'snow-dominated' should they be excluded from your analysis?*
Groundwater recharge and thus groundwater baseflow as well depend not only on precipitation but also on temperature if precipitation falls as snow. As our analyses were performed on monthly scale, this is only relevant for catchments where temperatures are regularly below zero degrees for several weeks. In our study these are mainly the catchments in the Alpine foothills where snow accumulates over winter and smelts in spring/summer so that annual minimum flows occur in winter/spring. For the low mountain ranges of central Germany there are only few catchments in our study where annual low flows occur occasionally in winter/spring. As we discuss, the results for these "snow-dominated" catchments have to be interpreted carefully. In accordance with the comment of Mr. Sawaske we will add the changes for summer low-flows in these catchments to our analysis.

*Specific comments*
*P1, l10: replace reflexion with reflection*

*P2, l28: resp. is not a normal abbreviation to use in articles like this. If it is and abbreviation for respectively please re-write [see also p5, l28, p10, l9 and p13, l25]*
*P9, l9, replace Marchant and Bloomfield (2013) with Bloomfield and Marchant (2013)*
*P10, l5, delete comma after revealed*
Will be corrected.

*P2, l28: resp. is not a normal abbreviation to use in articles like this. If it is and abbreviation for respectively please re-write [see also p5, l28, p10, l9 and p13, l25]*
*P9, l9, replace Marchant and Bloomfield (2013) with Bloomfield and Marchant (2013)*
*P10, l5, delete comma after revealed*
Will be corrected.

---

## Author Response (AR1)

We would like to thank Spencer Sawaske and John Bloomfield for their feedbacks and constructive suggestions on this manuscript. We think that our revisions based on the comments improved our work. Below we give point-by-point responses to the comments (blue and italic). Sections we changed/added in our new manuscript are marked in red.

5  S. Sawaske:

*General Comments*

*This is a sound and timely paper that builds on established methods and previous work of others and examines important water resource issues with management implications in the face of climate change. There is a*
10  *thorough introduction and discussion of methods. The analysis is based on a very large sample size of existing gages which strengthens the validity of the results and conclusions. Methods used in the analysis are well documented and established procedures in the field. There is a detailed appendix and appropriate and easily interpretable figures. I particularly appreciated figure 5, a very compelling illustration of the impacts of start and end year on trend analyses.*

*Specific Comments*
*The study includes data from presumably a wide range of catchment types based on the spatial distribution of gages. However, the only catchment attribute discussed is a measure of porosity. I would have preferred some greater detail of catchment geology, land use/vegetation, and geomorphic characteristics to differentiate the*
20  *study basins.*
Indeed, there is a larger number of potential controls that influence hydrologic processes on catchment scale. Here we chose those factors that are most likely to influence long term changes in minimal baseflow. We differentiated only three hydrogeological classes to maintain a sufficient sample size and this appeared the most straightforward classification to explore first-order controls. There is of course more lithological diversity
25  across Germany. One possible predictor would be the hydraulic conductivity. There are some estimates covering the whole study area, but uncertainty is very high. For example, the estimates of GLHYMPS (Gleeson et al., 2014) and HÜK200 (BGR and SGD, 2016) differ up to six orders of magnitudes for many parts of Germany. Classifying more detailed meaningful categories with enough samples each, would be a major ground work required and thus beyond the scope of this paper. We discuss this issue in more detail in the new manuscript:

The large differences of baseflow response times for different porosity classes match the theoretical assumptions that baseflow strongly depends on hydrogeological conditions. For the entire streamflow, differences were found to be much smaller (not shown, compare e.g. Haslinger et al., 2014), since other processes like overland flow are also important. The patterns of $T_R$ which are related to the catchments'
35  hydrogeology support the assumption that baseflow can be used as a proxy of the groundwater situation on catchment scale. *More detailed information on hydrogeology has the potential to further improve the understanding of differences in baseflow response. However, this information is not yet detailed enough for the scale and distribution of headwater catchments. For example, two large-scale estimates on hydraulic conductivity available for Germany (GLHYMPS from Gleeson et al. (2014) and HÜK200 from BGR and SGD*
40  *(2016)) were considered and found to differ up to six orders of magnitudes.*

Moreover, we added land use categories and a topography-based measure of potential storage as further characteristics to differentiate study basins. Both catchment factors are not related to observed trend magnitudes. These results are shown in Figure 6, we added Subfigures f) and g):

[Figure]

*I don't fully understand the use of the "scenario-neutral" term. Although emissions specific scenarios are not discussed, only one trajectory is analyzed (reduced summer precip. and increase winter precip.). It would be interesting to test the baseflow impacts of varied changes in seasonal precipitation pattern.*

We agree that other scenarios could be tested easily. Instead we opted to test only a generalized probable future change, as all projections agree on this general scenario (reduced summer precipitation and increased winter precipitation). The term scenario-neutral was branded first by Prudhomme et al. (2010) for tests that do not follow strict climate model runs, but we agree that we deviate from their more substantial sensitivity test/response surfaces and hence changed the terminology. However, as our test scenario is independent from the magnitude of change which depends on the model and emission scenario, the analysis is in a way "scenario neutral".

It is not in the scope of our study to identify changes due to theoretical combinations of precipitation changes but to predict the changes under certain assumptions that are common to all scenarios. Therefore, we decided not to include further scenarios.

*How are summer low flows in the Alpine foothills projected to change? Although minimum flows currently occur in winter, low flows in summer can have greater water resources and ecological implications.*

This is an important point and we included the implications for summer low-flows in this area in the revised version. Due to the short response times most catchments in the Alpine foothills are expected to have reduced summer low-flows under the test scenario.

According to the test scenario, $Q_{b7}$ will decrease or not change for most of Germany (Figure 7). The only catchments with estimated increases of $Q_{b7}$ are located in the mountainous regions of south-east Germany, especially in the Alpine foothills in the catchments with annual low-flows in winter. *However, also in the catchments with annual low-flows in winter or spring, impacts might be more severe for low-flows in summer or autumn. Additional tests revealed that for 22 (27) out of 35 catchments with $M_{Qb7}$ in winter or spring, baseflows of the summer (autumn) season are expected to decrease according to the test scenario.*

*It was important to note in section 5.2 that: "Catchments' characteristics response times were assumed to remain constant". Although likely outside the scope of this article, it would be nice to explore this more and the implications on baseflow recession.*

For climate change impact studies, the choice of boundary conditions is often critical. Catchment characteristics (e.g. response times) can generally be assumed to be independent from climate. However, recent studies found that over a longer term these catchment characteristics are also related to climate and so climate change might impact them as well (e.g. Troch et al., 2015; Saft et al., 2016). This is often referred to as catchment coevolution. Unfortunately, complexity of these processes is high and therefore not easily predictable for a large range of conditions like they occur in Germany. Therefore, we assume constant response times but agree, that this would be an interesting topic for further research. We expanded this point in our discussion:

Catchments' characteristic response times were assumed to remain constant. Under a more extreme climate change however, changes in catchments´ responses (e.g. due to non-stationary response times) cannot be excluded. Based on assumed precipitation change in the catchments' respective recharge periods, decreasing $Q_{b7}$ (i.e. increased groundwater drought hazard) were estimated for the majority of the catchments. This is because of the timing of $M_{Qb7}$, which occurs predominantly in summer/autumn. Only those catchments with long response times can compensate the decreases in summer precipitation and only in the Alpine foothills, where $M_{Qb7}$ is in winter, a decrease in drought hazard is predicted. However, in this region winter precipitation

is mainly snow and therefore does not immediately contribute to groundwater recharge. Moreover, air temperature and its changes over time – which are not considered in the test scenario – are especially important for winter low-flows. Hence, uncertainty is quite high for these projections. *If response times are not assumed constant, further conclusions may be drawn. A decrease of response times might be caused by decreased storage, e.g. due to urbanization, or accelerated drying and increased evapotranspiration in spring. Such changes would increase the drought hazard for most parts of Germany due to the timing of $M_{Qb7}$. An increase in response times by the opposite processes is possibly less likely but could compensate the reduced precipitation during summer for the catchments that are currently estimated to have increased drought hazard.*

*Technical Corrections*
*5.2, line 1 should be dependent rather than depending.*
Done.

*This is an easily readable and well formatted text.*

J. Bloomfield:

*The paper presents an analysis of baseflow 'to characterize groundwater drought on a catchment scale'. Trends in observed baseflow minima and derived drought descriptors are identified and investigated in the context of 'climatic and catchment controls'. A 'scenario-neutral' approach is adopted to characterise the sensitivity of the drought descriptors to future changes in. The study uses flow data from 338 gauges in headwater catchments across Germany.*

*General comments*
*Introduction:*
*The aims of the study could be set out more clearly in the Introduction. The two following statements 'we use a baseflow approach to characterize groundwater drought on a catchment scale' (p2, l19) and 'Employing a data-based approach, in this study we assess future changes in drought hazard on catchment scale across Germany' (p3, l29) describe what has been done, but there is no unambiguous aim or research question stated in the Introduction.*
Thanks for pointing this out. We set the main aim and the more specific objectives more clearly in the new manuscript:
*This study aims to explain differences in drought trends by catchment characteristics to allow for more accurate predictions under climate projection uncertainty on a headwater scale.* First, past trends in baseflow drought and catchment-relevant response times are analysed. Secondly, past trends are attributed to climatic and catchment controls. Finally, based on these statistics an estimate for future changes in baseflow drought valid for all common emission scenarios and climate models is realised.

*Study area and data:*
*Data from 338 gauges on headwater catchments were used in this study. In this context, what constitutes a headwater catchment?*
The term "headwater catchment" was chosen as all catchment areas are below 200 km². We state this more clearly in the revised version:

The dataset used in this study is the same set of headwater catchments *(that is all catchment areas are below 200 km²)* that were used in Hellwig et al. (2018) to evaluate the representativeness of meteorological grid data.

*It's not easy to tell from Figure 2, but it is possible that some of the gauges are nested catchments. Is this the case? If so, what biases if any might this introduce into the data? What are the implications of those potential biases for the trend analysis (section 4.2) and the results of the potential future drought hazard assessment (section 4.3)?*

We agree that nested catchments would introduce biases and hence did not use any nested catchments in this study. We added this information in the revised version.

The final dataset consisted of 338 *not nested and independent* gauges across Germany.

*Streamflow data for the period 1970-2009 was analysed. Was the data complete? If there was any missing data how was it accounted for in the pre-processing of the data?*

As we report in section 2, any records with data gaps were not considered for this study.

*The data 'were visually screened for signs of anthropogenic influence' and 'four of the gauges showed spurious changes … and were subsequently removed' (p4, l7-8). Please could you justify their exclusion in more detail. What anomalies were present in the data that caused you to exclude the sites?*

For all records, summed precipitation was plotted over summed streamflow for the entire period. Under constant conditions, the plot is expected to show seasonal varying slopes but no sudden knee. However, four of the gauges had a knee, indicating some kind of anthropogenic influence. In a Master thesis, L. Gerke was able to relate the knee in the plot of one catchment to a change of the gauges' location. We include these details in the new manuscript.

Even though the dataset consists of near-natural headwaters with minimal regulations, *the hydrographs and their double mass curves* were visually screened for signs of anthropogenic influence. *In a natural catchment under relatively constant conditions, the precipitation-streamflow relationship should be similar for the entire observation period.* Four of the gauges showed sudden changes in the relationship and were subsequently removed.

*Why was a 2/3rds fraction used to define mixed catchment (p4, l21)? Is there a citation for this?*

The fraction is arbitrary and was chosen to get the relatively homogenous groups "fractured" and "porous". Due to the high heterogeneity of the catchments we think it is hard to define more detailed groups with sufficient sample size. Using another reasonable fraction does not change the results (cf. figure below) and only few catchments are classified into another group. Changes in significance are negligible.

[Figure]

We added this information to the manuscript.

The catchments were classified according to the main type of porosity found in the underlying geology, either "porous" for porous aquifers such as unconsolidated alluvial and glacial fillings or "fractured" for fractured bedrock. If less than 2/3 of the catchment's area is covered by one of these types, the class "mixed" was assigned, including catchments which are dominated by other types of porosity like karst. *Additional analysis revealed that other reasonable thresholds to classify the catchments do not change the results.*

*Is there any information on the distribution of low permeability superficial deposits across the region? It's not uncommon for such deposits to play an important role in stream flow generation and so the proportion of such deposits in catchments might be an interesting parameter to investigate in the context of the study, and should at least be commented on, either here or in the Discussion.*

We don't know of data on superficial deposits across the region that are detailed enough to use them as predictors. We added this point to the discussion:

*More detailed information on hydrogeology has the potential to further improve the understanding of differences in baseflow response. However, this information is not yet detailed enough for the scale and distribution of headwater catchments. For example, two large-scale estimates on hydraulic conductivity available for Germany (GLHYMPS from Gleeson et al. (2014) and HÜK200 from BGR and SGD (2016)) were considered and found to differ up to six orders of magnitudes. Moreover, information on superficial deposits that might be particularly important for streamflow generation could enhance results but is not in detail available yet.*

*Methods:*

*The justification for the use of Mann-Kendall (MK) test could be more robust. There is a significant literature on the application of this test to hydroclimatic time series. However, it's application to such time series is also contentious when there are underlying auto and cross-correlations present. Consider adding to the justification of use of the zstatistic [also see comment below about identification of significant trends based on MK test].*

*Results:*

*Figure 4c is a map of the MK z-statistic indicating the direction and magnitude of the trend in Qb7. However, it is stated that 'according to the MK-Test, 40 out of the 338 catchments show a significant trend in Qb7' (p8, l7). What was the level of significance that was used? Please justify this and link this justification back to the Methods, section3.2?*

The Mann-Kendall test is a common test for statistical analysis of time series in hydrology. The detailed description of the method applied is given in Appendix B of the manuscript. Since MK overestimates significance for autocorrelated time series, we correct in this study for serial correlation using pre-whitening (see equation A6 in the manuscript). As we think that the detailed procedure is only interesting for part of the readers, we prefer to leave most of this information in the Appendix, however we agree that the significance level ($\alpha=0.05$) should be mentioned in the main text as well and added this point.

The analysis was carried out with the non-parametric Mann-Kendall test (MK), which is a common tool to detect monotonic increases or decreases in hydrological time series (e.g. used by Douglas et al. 2000, Lins and Slack 1999, Lorenzo-Lacruz et al. 2012, Rennermalm et al. 2010, Asarian et al., 2016) and is robust to outliers. The test was performed after pre-whitening the time series to account for the influence of serial correlation (see Appendix B for calculation steps). *A significance level of $\alpha = 0.05$ was used.*

*Discussion:*

*It is noted in Section 5.1 of the Discussion that 'a baseflow approach does not allow for conclusions on groundwater storage in snow-dominated catchments' (p9, l30). In this context, how do you define snow-dominated catchments, and which if any of the 338 headwater catchments that you have analysed fall into this category? If any of the catchments are 'snow-dominated' should they be excluded from your analysis?*

Groundwater recharge and thus groundwater baseflow as well depend not only on precipitation but also on temperature if precipitation falls as snow. As our analyses were performed on monthly scale, this is only relevant for catchments where temperatures are regularly below zero degrees for several weeks. In our study these are mainly the catchments in the Alpine foothills where snow accumulates over winter and melts in spring/summer so that annual minimum flows occur in winter/spring. For the low mountain ranges of central Germany there are only few catchments in our study where annual low flows occur occasionally in winter/spring. As we discuss, the results for these "snow-dominated" catchments have to be interpreted carefully. In accordance with the comment of Mr. Sawaske we added the changes for summer low-flows in these catchments to our analysis (cf. our answer to the third comment of S. Sawaske).

*Specific comments*
*P1, l10: replace reflexion with reflection*
*P2, l28: resp. is not a normal abbreviation to use in articles like this. If it is and abbreviation for respectively please re-write [see also p5, l28, p10, l9 and p13, l25]*
*P9, l9, replace Marchant and Bloomfield (2013) with Bloomfield and Marchant (2013)*
*P10, l5, delete comma after revealed*
Done.

[revised manuscript text omitted]